# Acute Exercise and the Systemic and Airway Inflammatory Response to a High-Fat Meal in Young and Older Adults

**DOI:** 10.3390/metabo12090853

**Published:** 2022-09-10

**Authors:** Stephanie P. Kurti, William S. Wisseman, Molly E. Miller, Hannah L. Frick, Steven K. Malin, Sam R. Emerson, David A. Edwards, Elizabeth S. Edwards

**Affiliations:** 1Human Performance Laboratory, Department of Kinesiology, James Madison University, Harrisonburg, VA 22807, USA; 2Morrison Bruce Center, James Madison University, Harrisonburg, VA 22807, USA; 3Department of Kinesiology and Health, Division of Endocrinology, Metabolism and Nutrition, Rutgers University, Piscataway, NJ 08854, USA; 4Department of Nutritional Sciences, Oklahoma State University, Stillwater, OK 74078, USA; 5Department of Kinesiology, University of Virginia, Charlottesville, VA 22904, USA

**Keywords:** post-prandial, aging, inflammation, airways, acute exercise, Western diet

## Abstract

The purpose of the present study was to determine fasting and high-fat meal (HFM)-induced post-prandial systemic inflammation and airway inflammation (exhaled nitric oxide (eNO)) in older adults (OAs) compared to younger adults (YAs) before and after acute exercise. Twelve YAs (23.3 ± 3.9 y *n* = 5 M/7 F) and 12 OAs (67.7 ± 6 y, *n* = 8 M/4 F) completed two HFM challenges. After an overnight fast, participants underwent an HFM session or pre-prandial exercise (EX, 65% VO_2Peak_ to expend 75% of the caloric content of the HFM) plus HFM (EX + HFM) in a randomized order. Systemic inflammatory cytokines were collected at 0, 3, and 6 h, while eNO was determined at 0, 2, and 4 h after the HFM (12 kcal/kg body weight: 61% fat, 35% CHO, 4% PRO). TNF-α was higher in OAs compared to YAs (*p* = 0.005) and decreased across time from baseline to 6 h post-HFM (*p* = 0.007). In response to the HFM, IL-6 decreased from 0 to 3 h but increased at 6 h regardless of age or exercise (*p* = 0.018). IL-8 or IL-1β did not change over the HFM by age or exercise (*p* > 0.05). eNO was also elevated in OAs compared to YAs (*p* = 0.003) but was not altered by exercise (*p* = 0.108). There was a trend, however, towards significance post-prandially in OAs and YAs from 0 to 2 h (*p* = 0.072). TNF-α and eNO are higher in OAs compared to YAs but are not elevated more in OAs post-prandially compared to YAs. Primary systemic inflammatory cytokines and eNO were not modified by acute exercise prior to an HFM.

## 1. Introduction

The typical Western diet consists of nutrient-poor and calorically dense foods, that are high in saturated and trans fats as well as sugar. In addition, individuals may consume multiple meals a day, placing them in a post-prandial state for most of the day [1]. The negative health effects of chronic high-fat meals (HFMs) span from metabolic dysfunction to pulmonary complications that include, but are not limited to: obesity, type 2 diabetes, cardiovascular disease (CVD), and respiratory diseases such as asthma [2,3]. A key mechanism for the development of such lifestyle-related diseases is the activation of the immune system with persistent low-grade inflammation [4]. While HFM diets are associated with chronic low-grade inflammation, even an acute HFM increases systemic and airway inflammation [5,6], though results are conflicting [7,8,9]. Considering inflammation is a hallmark characteristic of CVD, identifying who may be most susceptible to deleterious changes after HFM consumption and how to best treat inflammation are paramount for improving medical care. 

In young and middle-aged adults between the ages of 18–60 years, a recent systematic review demonstrated that of the commonly assessed pro-inflammatory cytokines, IL-6 most consistently increased post-HFM, on average by 100% (~1.5 pg/mL to 2.9 pg/mL) 6 h post-meal. In contrast, TNF-α, IL-1β, and IL-8 were more variable [7]. While findings for systemic changes in inflammation seem inconsistent, the increase in airway inflammation after a single HFM is more consistent. Indeed, we and others have demonstrated in healthy, non-asthmatic young adults that exhaled nitric oxide (eNO) levels increase at 2 h post-HFM [10,11,12,13,14] and sputum neutrophils rise 4 h post-HFM [15]. A gap, though, in the literature on systemic and airway inflammation is that most work to date has been conducted in young adults, and research on the post-prandial response in older adults is limited. Considering older adults are at greater risk for developing CVD, type 2 diabetes, and pulmonary diseases, as well as are more susceptible to age-related deleterious immune changes which can elevate systemic and airway inflammation [16], understanding post-prandial responses in this population is important. Therefore, we tested the hypothesis that post-prandial systemic and airway inflammation was higher in older adults (OAs) compared to younger adults (YAs). Secondarily, we previously showed that exercise was effective at attenuating post-prandial lipemia and glycemia older adults [17], but whether this change in circulating substrates is related to inflammation is unclear. We thus examined whether an acute bout of pre-prandial exercise modified airway and systemic inflammation in these cohorts. 

## 2. Materials and Methods

Twelve YAs (23.3 ± 3.3 years, *n* = 5 M/7 F) and 12 older adults (OAs) (67.7 ± 6.0 years, *n* = 8 M/4 F) participated in this study. The methods employed in this study were part of a larger study investigating post-prandial glycemia, lipemia, and metabolic load index [17]. These were the same participants, and some data are reported here for ease of interpretation, and participant demographics are included in Table 1. Participants signed an informed consent and then completed a physical activity readiness questionnaire (PAR-Q). The participants were free of CVD, type 2 diabetes, or pulmonary diseases and were not on any medications known to interfere with any outcomes assessed in the present study. All procedures were approved by the James Madison University Institutional Review Board in accordance with the *Declaration of Helsinki*. 

### 2.1. Experimental Design

Each participant completed a total of 4 laboratory visits. On the first visit, initial baseline measures of height, weight, body fat percentage, and blood pressure were taken. The participants then performed an incremental test to volitional exhaustion to determine peak oxygen consumption (VO_2Peak_). Participants were then randomly assigned to perform either the HFM alone or HFM + exercise (EX + HFM; Figure 1). Participants began each meal trial in the morning (starting time: 5:00–9:00 a.m.) after a 12 h overnight fast with identical starting times within participants. There was a 6 h post-prandial period after consumption of the HFM, where cytokines were assessed at 0, 3, and 6 h and eNO at 0, 2, and 4 h post-prandially.

### 2.2. Initial Visit 

Height was taken with a stadiometer (Charder Model HM 200P, Charder Electronic Co Ltd., Taichung, Taiwan) and weight with a standard physician’s scale (Dymo Pelouze model 4040, Newell Brands, Hoboken, NJ, USA) to calculate body mass index. Dual-energy X-ray absorptiometry (DXA) (GE Lunar iDXA Fairfield, CT, USA) was also performed to calculate lean mass and body fat. Resting blood pressure (BP) was taken twice using an automatic sphygmomanometer (ProBP 3400 Welch Allyn, Skaneateles Falls, NY, USA) after the DXA with at least 5 min of rest in the seated position. The average of the two BP measurements was recorded. 

An incremental cycling test to volitional exhaustion was performed on a VIAsprint 150P cycle ergometer (Vyaire Medical, Mettawa, IL, USA) to determine VO_2Peak_ (Table 2). Breath samples were analyzed by a Vmax Encore metabolic cart (Vyaire Medical, Lake Forest, IL, USA), while heart rate (HR) was measured by a Polar heart rate strap and watch (Lake Success, NY, USA). Participants completed a 5 min warm-up at a self-selected cadence of ≥50 revolutions per minute (rpm). During the warm-up, power output was either increased or decreased every minute until participants identified a workload that they perceived as sustainable for approximately thirty minutes. After this, the incremental test began, and resistance was increased every minute by 10 or 25 Watts as appropriate as described before [17]. The test was terminated when participants reached volitional exhaustion. 

### 2.3. Pre-Prandial Exercise Bout

EX was performed 12 h prior to their HFM challenge on the VIAsprint 150P cycle ergometer (Vyaire Medical, Mettawa, IL, USA). Wattage was set so that the participants’ HR stayed within an HR range that corresponded to 60–70% of their VO_2Peak_ HR. The exercise duration was personalized to achieve a total energy expenditure equivalent to 75% of the calories of the HFM challenge (Equation (1) shown below). However, participants were not permitted to exercise longer than 2 h. There was no caloric replacement after the exercise bout due to the potential impact of partial caloric replacement on post-prandial lipemia (PPL) [18].

Equation (1): (1)$$\frac{mass\;in\;kg*12.5\;kcals*0.75}{absolute\;VO2*4.825L\;O2/kcal*0.65}$$

### 2.4. HFM Protocol

After the overnight fast, participants reported to the lab and were seated in a reclining phlebotomy chair. The HFM challenge consisted of a slice of Marie Callender’s Chocolate Satin Pie (Conagra Brands, Chicago, IL, USA) portioned out to a caloric density of 12 kcal/kg body weight. Macronutrient composition of the HFM challenge was 61% fat (58% saturated fat), 4% protein, and 35% carbohydrate. After reporting to the laboratory, participants were asked to remain sedentary for the duration of the HFM challenge session, except when inside the restroom. In the HFM challenge alone condition, participants were instructed to refrain from exercise for at least 48 h prior to their visit. After five minutes, two baseline BP measurements were taken with at least 30 s of rest between each recording. An indwelling safelet catheter was inserted into a forearm vein via a 22-gauge needle (Fisher Scientific, Waltham, MA, USA) and kept patent with a 0.9% NaCl solution. Blood was then collected at 0, 3, and 6 h and eNO at 0, 2, and 4 h. Timing for the assessment of systemic inflammatory cytokines and eNO was performed when our group and others have shown these markers are at or near their peak levels. Future work nonetheless should consider more time-points to depict time-course kinetics across age, with and without exercise.

### 2.5. Cytokine Analysis

Blood was immediately centrifuged for 12 min at 1800× *g* at 4 degrees Celsius. Plasma was then aliquoted into 0.5 mL microtainers and stored in a freezer at −80 degrees Celsius. Plasma samples were assayed at Eve Technologies (Calgary, AB, Canada). A Human High Sensitivity T-Cell Discovery Array 14-plex (HDHSTC14) was used to measure primary outcomes of IL-6, IL-1β, IL-8, and TNF-α. The major pro-inflammatory cytokines are the primary focus of this paper, while the remaining cytokines that were assessed are IFN-γ, GM-CSF, IL-2, IL-4, IL-5, IL-10, IL-12, IL-13, IL-17, and IL-23. Biomarkers were analyzed in duplicate, and the average measurement was used for analysis. 

### 2.6. Airway Inflammation

Participants were instructed to sit up straight with their feet flat on the floor and without a nose-clip in order to collected eNO as an indicator of airway inflammation [19]. Participants deeply inhaled while on the mouthpiece and then performed a steady exhale that lasted approximately 6 s. Testing was performed 2 times, with measurements within 5% of one another, and the average was used for analysis. eNO was analyzed via chemiluminescence on a Niox Vero Analyzer (Circassia, Lake Park, IL, USA) according to American Thoracic Society guidelines [20].

### 2.7. Statistical Analysis

Statistical analyses were performed with IBM SPSS Statistics v26.0 (IBM Corporation, Armonk, NY, USA) and GraphPad Prism (Graphpad software, Inc., San Diego, CA, USA). All data were analyzed for normality using the Shapiro–Wilk test. Considering baseline systemic and airway inflammation passed the Shapiro–Wilk test of normality and inspection of central tendency, a three-factor analysis of variance was performed with the within-participants factor of time (0, 3, and 6 h for cytokines; 0, 2, and 4 h for eNO) and between-participants factors of condition (EX or NE) and age (YA and OA). If significant main effects of group or time were detected, data were assessed to determine significance as a linear or quadratic function by contrast analysis. Significance was set to *p* < 0.05. Data are mean (SD). 

## 3. Results

### 3.1. Participant Characteristics and Exercise Sessions

Participant demographics and maximal/submaximal exercise have been published previously but are shown here for ease of interpretation (Table 1 and Table 2; 18). There were no significant differences in participant demographics between OAs and YAs except for height (*p* = 0.02). The OAs and YAs had similar kilocalorie consumption of the HFM (968.8 ± 180.9 and 856.8 ± 207.1, respectively) and exercise duration and energy expenditure in the exercise condition (all, *p* > 0.05). There were also no differences in absolute VO_2peak_ (*p* > 0.99, Table 2), so differences in relative VO_2peak_ were due to differences in body mass between the OAs and YAs. 

### 3.2. Systemic Markers of Inflammation: Primary Outcomes

Primary outcomes for cytokines (IL-6, IL-8, TNF-α, IL-1β) are displayed in Figure 2A–D. Additionally, outcomes with significant results (IFN-*γ*, IL-2, IL-13, and IL-17) are also displayed on Figure 2E–H. Each cytokine will be hereafter described with main effect across time-points, between the EX and no exercise (NE) condition, between YAs and OAs, and then whether there were any interaction effects. IL-6 was significant across time (*p* = 0.018), where it decreased at 3 h and returned to baseline at 6 h, but there were no interaction effects (*p* > 0.05) or differences between age (*p* = 0.809) or condition (*p* = 0.809). There were no significant findings by time (*p* = 0.453), age (*p* = 0.141), or condition (*p* = 0.818) for IL-8, and no significant interaction effect (*p* > 0.05). TNF-α decreased over the 6 h (*p* = 0.007), but there was no difference between the EX and NE (*p* = 0.074). There was a significant difference by age (*p* = 0.005), where the OAs had higher TNF-α compared to the YAs. There were no significant interaction effects between time, condition, and age (all *p* > 0.05). There was no significant effect across time-points for IL-1β (*p* = 0.321), by age (*p* = 0.392), or between conditions (*p* = 0.880), and no interaction effects between any variables (all *p* > 0.05).

### 3.3. Secondary Cytokine Outcomes

There were no significant effects of GM-CSF across time (*p* = 0.847), condition (*p* = 0.423), and age (*p* = 0.998) or an interaction effect between time, condition, and age (all *p* > 0.05). IFN-*γ* did not change across time (*p* = 0.647), however, OA-NE and OA-EX had a lower response compared to YA-EX and YA-NE (*p* < 0.001). There were no significant interaction effects between time, condition, and age (all *p* > 0.05). There were also no significant changes across time points for IL-10 (*p* = 0.468), between EX and NE (*p* = 0.959), or between OAs and YAs (*p* = 0.647). There were no significant interactions effects between time, age, and condition (all *p* > 0.05). IL-2 was significantly different by age, where the OAs had lower IL-2 compared to the YAs (*p* = 0.002), however, there were no effects of any other variables (all *p* > 0.05). There was a trend towards significance for IL-5 by age (*p* = 0.084), where the OAs had a higher response that the YAs. However, there were no interaction effects (all *p* > 0.05) or a difference across time (*p* = 0.637) and by condition (*p* = 0.889). There were no significant interaction effects or effect of time, condition, and age for IL-12 or IL-23 (all *p* > 0.05). There was a significantly lower response in IL-13 by age where the OAs had lower levels of IL-13 compared to YAs (*p* = 0.049) but no difference between EX and NE (*p* = 0.736). There was a significantly higher response in IL-17 in the OAs compared to the YAs (*p* < 0.001), but no difference between the EX and NE conditions (*p* = 0.362). There were no significant interaction effects between age, condition, and time (all *p* > 0.05).

### 3.4. eNO

The post-prandial eNO results are displayed in Figure 3. There were no significant interaction effects between time, condition, and age. There was no significant difference in eNO across time from baseline to 2 and 4 h as a main effect of time (*p* = 0.944), however, it was trending towards significance as a quadratic function where there was an increase from baseline to 2 h, which returned near baseline values at 4 h (*p* = 0.072). There were no significant differences in the eNO response by condition (*p* = 0.180), however, eNO was significantly higher in the OAs compared to YAs (*p* = 0.003).

## 4. Discussion

Determining who may be most susceptible to changes in inflammation is critical since inflammation is present in pulmonary and metabolic aliments and CVD. Additionally, identifying whether exercise may potentially mitigate post-prandial inflammation and ultimately chronic disease risk is an important area for clinical application and possibly treatment. The primary aims of this study were to determine whether aging would elevate post-prandial systemic and airway inflammation, as well as to elucidate whether an acute bout of exercise could reduce systemic and airway inflammation in older and younger adults. Our hypotheses were partially supported in that several pro-inflammatory markers of systemic inflammation, such as TNF-α, were elevated in OAs compared to YAs, while other markers such as IFN-**γ** were elevated in YAs compared to OAs. Our hypothesis that OAs would exhibit higher eNO values compared to YAs was supported by the data too, though the magnitude of increase in post-prandial airway inflammation was the same in OAs and YAs. There were also no changes in post-prandial systemic inflammation or eNO with acute exercise, albeit a trend in TNF-α reductions.

### 4.1. The Impact of Age and Exercise on Systemic Inflammation

Considering inflammation is associated with a wide array of chronic diseases, determining whether OAs have elevated inflammation in the fasting and/or post-prandial state is relevant. The cytokines IL-6, TNF-α, IL-8, and IL-1β have been shown to rise in the post-prandial period, though results are conflicting, as previously mentioned [7]. IL-6 is the most frequently reported acute phase cytokine. In the current study, there was no change in IL-6 from 0 to 6 h, and the decrease at 3 h was consistent with prior findings from previous studies [21]. In fact, Emerson et al. utilized the same HFM protocol (Marie Callendar’s Chocolate Satin Pie at 12 kcal/kg BW) and observed a similar quadratic response for IL-6 in older adults, with a decrease at 3 h and a return to baseline at 6 h. This finding was similar to data from Teeman et al., who used an HFM challenge of 10 kcal/kg BW and 63% fat and reported decreases in IL-6 across time-points from baseline to 4 h post-HFM [22]. However, additional work suggests post-prandial increases in IL-6 even with similar or smaller meal sizes (from 500–760 kcals and 59% fat [7,23]). One proposed mechanism for the differences in post-prandial IL-6 across the literature is based on the diurnal changes in IL-6. Concentration of IL-6 increases upon awakening and declines throughout the day. As the HFM sessions in the present study were all performed in the morning after a prolonged overnight fast, diurnal changes may explain the decrease in IL-6 at 3 h post-prandially [24,25]. However, protocols involving morning HFMs are common even in experiments that have reported increases in post-prandial IL-6 from baseline measurements. Additionally, baseline IL-6 levels in the present study were not higher than commonly reported in the literature (~1.0 pg/mL in present study) [7], though baseline levels of IL-6 can be variable between groups based on physical activity status and other factors [26,27].

TNF-α is one of the most frequently measured inflammatory markers regarding post-prandial inflammation, although Emerson et al. suggest it rarely changes in the post-prandial state [7]. There does not appear to be a specific HFM fat percentage, caloric load, or participant characteristic which consistently elicits an increase in TNF-α, yet there appears to be some correlations that exist after an HFM in post-prandial TNF-α and IL-6 responses. Interesting, with increases in IL-6 there are reported increases in TNF-α post-HFM [28], yet with no increase in IL-6 decreases in both TNF-α and IL-1β were reported [21,29]. Herein, we observed the latter, highlighting that regulation of IL-6 and TNF-α can function under different regulatory steps. In addition, since TNF-α decreased post-prandially, this may explain why there were no downstream increases in pro-inflammatory markers IL-8 and IL-1β since TNF-α is considered a master regulator of inflammation in many diseases [30]. Either way, it is interesting that there was a significant decrease in TNF-α in both groups, even though OAs had elevated levels compared to YAs. Aging alone, irrespective of acute exercise, can increase inflammation [16], particularly with older age [1]. Previous findings in TNF-α align with the current study since it was ~23% higher in OAs at baseline compared to YAs. This observation is clinically relevance since high TNF-α raises the risk for lifestyle-related diseases [31,32].

Interestingly, in the current study anti-inflammatory IL-10 decreased at 3 h post-meal compared to baseline. This is consistent with prior work during the post-prandial period [22]. It has been suggested that the decrease in IL-10 may be explained by declines in IL-6, though the decrease in IL-10 was not significant in the present study. Although IL-6 may be released as a pro-inflammatory cytokine related to increased risk of CVD [33], it can serve as an anti-inflammatory myokine secreted from skeletal muscle with exercise [34]. Considering IL-6 upregulation with exercise may increase IL-10 downstream [35], the decrease in IL-6 observed here in the present study may partially explain the lack of any changes in anti-inflammatory IL-10.

### 4.2. Post-Prandial Airway Inflammation, Aging, and Exercise

Findings from the present study are consistent with previous literature showing increased eNO in OAs compared to YAs [36]. Surprisingly, even with increased eNO at baseline in OAs, post-prandial eNO was not different between groups. However, it was trending towards significance with an increase at 2 h when it is typically seen post-prandially. It is possible that no differences in eNO post-meal were observed because of the lack of significance in post-prandial systemic inflammation, though cytokines do not need to increase for significant increases in eNO to be reported [11]. As previously stated, eNO increases following an HFM at approximately 2 h and returns to baseline around 4 h post-HFM [11,14]. However, the direct mechanisms remain unclear. In fact, it appears an HFM may impact the airways more than systemic inflammation [6], and that the increase may come from lipopolysaccharides (LPSs), a component of saturated fatty acids found in HFM. LPS increases Toll-like receptor (TLR) activation that has an independent and dependent pathway. The TLR4-independent pathway increases ROS and IL-1β levels through the NLRP-ASC pathway [37]. Pro-inflammatory cytokine IL-1β upregulates inducible nitric oxide (iNO), increasing eNO, while also decreasing insulin sensitivity to beget more inflammation [37,38]. In the present study, there were no increases in IL-1β post-prandially or differences between YAs and OAs, which may have been why there were lower increases in eNO than in our previous work. In either case, OAs in the current study increased eNO ~1.8 ppb with the HFM alone while the YAs increased ~0.9 ppb in the HFM alone condition. Previous work in healthy adults suggests that an acute increase in eNO may not be clinically significant since it does not impair pulmonary function, but is consistently statistically significant with absolute eNO increases between ~3.4 ppb and 6.5 ppb [11,12]. However, a similar absolute increase in eNO was seen in individuals with asthma, and was sufficient to impair bronchodilator recovery as well as pulmonary function [6]. If excessive inflammation is left untreated, tissue damage may occur and elicit structural changes to the bronchial walls which can lead to hallmark characteristics of asthma such as airway hypersensitivity and airway hyperresponsiveness [39].

Consistent with prior acute exercise work [11,14,15], our group has reported that acute exercise does not impact eNO responses after an HFM. Data from Scott et al. showed eNO is reduced after exercise by ~2.5 ppb in physically inactive adults with asthma [40]. Interestingly, the mean pooled eNO in our YAs and OAs without exercise was 22.4 ppb and in the exercise condition it was 19.3 ppb, with a mean change of ~3 ppb the morning after engaging in exercise. While this was not statistically significant (*p* = 0.18), it is worth mentioning that the absolute changes align with previous work in a clinical population and should be investigated further to determine whether these absolute changes confer increases in pulmonary function in clinical populations.

### 4.3. Experimental Considerations 

Other measurements of airway inflammation include more direct assessments such as induced sputum induction cell count and Toll-like receptor 4 mRNA expression by real-time PCR [6,15], or sputum gene expression confirmed with PCR (Li). Niox Vero has been shown to be valid and reliable [41], yet while the Niox Vero and Sievers have significantly high correlations with one another, some report that Niox Vero tends to report eNO values at about 30 lower than the Sievers [42], while others report the Niox Vero to report higher eNO levels [43]. While our primary interest was in assessing the change post-prandially, a reduction in the sensitivity of the device could have impacted the absolute change in eNO in the present study. Next, while some suggest hydration status may influence systemic inflammation, we did not assess hydration status. However, previous research suggests that changes in cytokine levels with exercise are not influenced by hydration [44]. Our goal was to match the groups by sex, however, enrolling OAs who met the inclusion criteria was challenging. While the small sample size could have limited our ability to assess our main outcomes, sub-analyses (data not shown) show no sex by time interactions for eNO or inflammation. Thus, sex is unlikely to have influenced these findings. In the present study, the OAs and YAs had similar absolute fitness levels, and higher fitness or chronic PA levels may lead to lower overall systemic inflammation [45]. Additionally, acute exercise may transiently increase or decrease cytokine levels [44], and physical activity status may alter the magnitude to which systemic inflammation increases post-exercise [46]. Exploration into the time course of cytokine changes after exercise and whether those changes could impact post-prandial inflammation would be an interesting follow-up study, but was outside the scope of the current study.

## 5. Conclusions

Several markers of systemic inflammation may be impacted by age, most notably TNF- α, which has been directly assessed and is increased in inflammatory pathologies such as rheumatoid arthritis and inflammatory bowel disease [47]. Still, an HFM that has been previously shown to elevate blood lipids and glucose does not appear to increase pro-inflammatory markers such as IL-6, IL-8, or IL-1 β. Additionally, eNO is higher in OAs compared to YAs and was trending toward significance in increasing post-prandially. When there is not an HFM-induced increase in eNO and systemic inflammation, neither cytokine levels nor airway inflammation appears to be altered with pre-prandial acute exercise. Further work should investigate both the physiological mechanisms behind changes in post-prandial cytokines and eNO and investigate whether reductions in pro-inflammatory cytokines or eNO confer health benefits in a clinical or older population.

## Figures and Tables

**Figure 1 metabolites-12-00853-f001:**
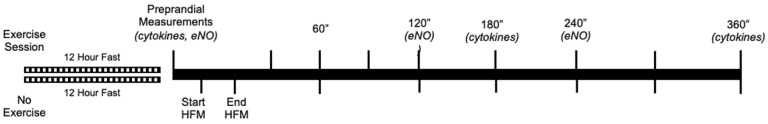
Experimental design in the EX + HFM or HFM alone session. A baseline blood draw was taken after a 12 h overnight fast for cytokine assessments and eNO was then performed. Cytokines were also assessed at 3 h (180 min) and 6 h (360 min) post-HFM while eNO was measured at 2 h (120 min) and 4 h (240 min) post-meal.

**Figure 2 metabolites-12-00853-f002:**
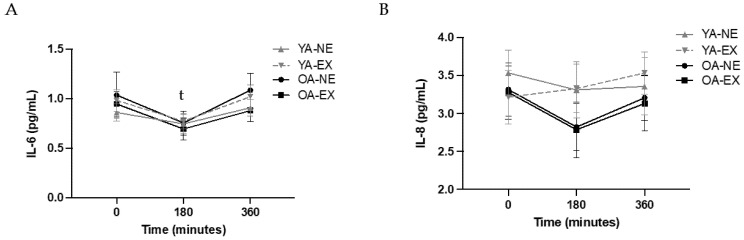
(**A**–**H**). Fasting and post-prandial cytokines from baseline to 360 min after the HFM. Concentrations were taken at baseline, 180 min, and 360 min post-HFM and analyzed with a three-way ANOVA for time, age (OA and YA), condition (EX and NE). Data are expressed as mean ± SEM. ^t^ Significant effect across time post-meal. ^a^ Significant impact of age, where OA is different from YA.

**Figure 3 metabolites-12-00853-f003:**
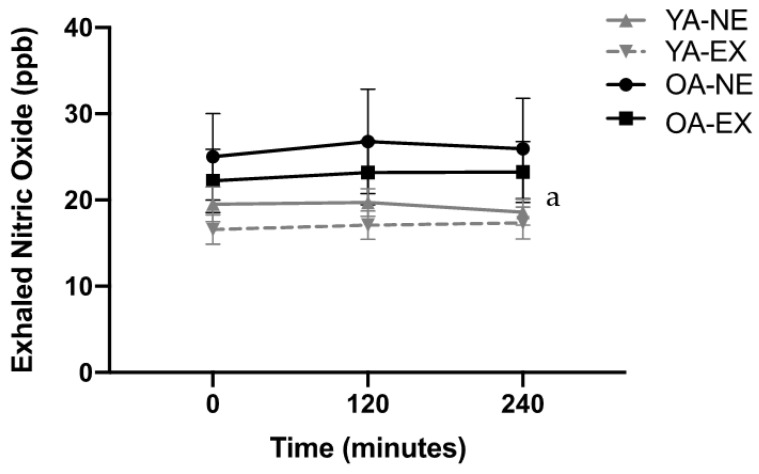
Exhaled nitric oxide from baseline to 240 min post-HFM. eNO was assessed during fasting, 120 min, and 240 min post-meal. Analysis was carried out with a three-way ANOVA for time, age (OA and YA), and condition (EX and NE). Data are expressed as mean ± SEM. ^a^ Significant impact of age, where OA is different from YA.

**Table 1 metabolites-12-00853-t001:** Participant Demographics.

	YA (*n* = 12/5 M, 7 F)			OA (*n* = 12/8 M, 4 F)		
	Mean		SD	Mean		SD
Age (years)	23.4	±	3.8	67.4	±	5.0 *
Height (cm)	167.5	±	8.1	176.0	±	8.9 *
Weight (kg)	71.4	±	17.3	80.7	±	15.1
Body Mass Index (BMI) (kg/m^2^)	25.3	±	5.0	25.8	±	3.1
Total Body Fat (%)	28.2	±	8.7	33.1	±	6.5
Android Body Fat (%)	30.1	±	13.1	40.3	±	9.7

* Significantly different from YAs (*p* < 0.05).

**Table 2 metabolites-12-00853-t002:** Exercise Data.

	YA (*n* = 12/5 M, 7 F)			OA (*n* = 12/8 M, 4 F)		
	Mean		SD	Mean		SD
VO_2peak_ (L/min)	2.3	±	0.4	2.3	±	0.8
VO_2peak_ (mL/kg/min)	33.4	±	5.3	28.3	±	6.7 *
Peak Power (Watts)	199.2	±	42.1	198.3	±	67.9
Peak Heart rate (bpm)	187.9	±	11.7	156.6	±	8.5 *
Exercise Duration (min)	88.1	±	15.2	105.6	±	25.7
Exercise Bout Expenditure (kcals)	642.6	±	155.4	726.7	±	135.6

* Significantly different from YAs (*p* < 0.05).

## Data Availability

Not applicable.

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
