# Peer review of "Acute Exercise and the Systemic and Airway Inflammatory Response to a High-Fat Meal in Young and Older Adults"

_metabolites, 2022, doi:10.3390/metabo12090853_

Round 1
Reviewer 1 Report
The paper is interesting , however it could be improved . The authors need to clarify that the acute and intense exercise itself can increase cytokines while a moderate exercise can reduce cytokines level , expecially if regularly practiced. In any case the hydrhation status can offers some points of reflexions and therefore the water distribution should be insert among the data . In the discussion session the authors need to clarify the clinical impact of the results and the eventual application in sports medicine expecially in older population .
Author Response
We thank this reviewer for pointing this out, and we were remiss to mention that chronic physical activity level and/or fitness level may alter cytokine levels, while acute exercise may transiently increase cytokines. This has been added to the experimental considerations on lines 398-404 which state,
“In the present study, the OA and YA had similar absolute fitness levels, and higher fitness or chronic PA levels may lead to lower overall systemic inflammation [45]. Also, acute exercise may transiently increase or decrease cytokine levels [44], and physical activity status may alter the magnitude to which systemic inflammation increases post-exercise [46] . Exploration into the time course of cytokine changes after exercise and whether those changes could impact postprandial inflammation would be an interesting follow-up study, but was outside the scope of the current study”.
Unfortunately, we do not have water distribution data. From our research, hydration status does not consistently appear to alter exercise-induced systemic inflammation or changes in cytokine levels (Svendsen et al. 2014). Therefore, while it would be interesting to explore the effect of hydration status if we had collected these data, we respectfully do not feel it is a necessary component to add since many other factors may impact systemic inflammation that are outside the scope of the current study (i.e. chronic dietary habits, genetic factors, other lifestyle factors such as psychological stress, etc.). Still, we have added this as an experimental consideration on lines 391-393 which state,
“Next, while some suggest hydration status may influence systemic inflammation, we did not assess hydration status. However, previous research suggests that changes in cytokine levels with exercise are not influenced by hydration [44]”.
Finally, we have added the clinical application to the discussion on lines 280-284 which read as follows,
“Determining who may be most susceptible to changes in inflammation is critical since inflammation is present in pulmonary and metabolic aliments and CVD. Also, identifying whether exercise may potentially mitigate post-prandial inflammation and ultimately chronic disease risk is an important area for clinical application and possibly treatment”.

Reviewer 2 Report
The study investigated the systemic and airway inflammatory response to a Western meal in young and older adults and whether an acute exercise bout impacts these outcomes. The study was well-carried and written. I have some concerns that needs to be addressed.
The participants of the study they were active physically?
In the abstract HFHC meal as being 57% fat, 39% CHO, and 4% PRO. In methods, the authors described HFHC meal as being 61% fat, 4% protein, and 35% carbohydrate. Please, clarify and standardize this information in a revised manuscript.
Please, add the fatty acid composition of the high-fat diet used in the study.
Why do the authors consider high-carbohydrate since 39% of calories were derived from carbohydrates? In my opinion, the authors used a high-fat low-protein diet. Please, clarify and justify this concern.
The authors analyzed several cytokines, but not all are shown in the figures. Please, correct it and present all results.
Lines 199-200: “. There were also no significant changes across time points for IL-10 (p = 0.468)”. However, in lines 322-323 the authors described “anti-inflammatory IL-10 decreased at both 3 and 6 hours compared to baseline in the current study”. Please, correct it.
Author Response
The participants of the study they were active physically?
In this study, we did not specifically recruit physically active participants since the primary comparison was by age (OA versus YA). Interestingly, OA and YA had similar fitness levels (VO2peak= 2.3 L/min in both YA and OA). We have clarified this study population in the results and experimental considerations per the reviewers’ comments on lines 180-182.
In the abstract HFHC meal as being 57% fat, 39% CHO, and 4% PRO. In methods, the authors described HFHC meal as being 61% fat, 4% protein, and 35% carbohydrate. Please, clarify and standardize this information in a revised manuscript.
We appreciate the reviewer for their attention to detail. The pie was 36 grams of fat per serving and 530 calories per serving (36 grams x 9 kcal/gram= 324 calories from fat). Therefore, the composition is 61% fat. We have edited the percentage of fat and carbohydrate in the revised manuscript.
Please, add the fatty acid composition of the high-fat diet used in the study.
The pie was 58% saturated fat which is now added on lines 133-134, and reads as follows,
“Macronutrient composition of the HFM challenge was 61% fat (58% saturated fat), 4% protein, and 35% carbohydrate”.
Why do the authors consider high-carbohydrate since 39% of calories were derived from carbohydrates? In my opinion, the authors used a high-fat low-protein diet. Please, clarify and justify this concern.
This is a great point. Originally, we were thinking that relative to other high fat meal studies in the literature, the percentage of carbohydrate in the present study is higher . However, since high carbohydrate is typically >45% of the meal, we have changed from high-fat, high-carbohydrate (HFHC) to high-fat meal (HFM) throughout the manuscript.
The authors analyzed several cytokines, but not all are shown in the figures. Please, correct it and present all results.
We have included all cytokine results either in text or in the figures. Since the primary outcomes were IL-6, IL-1β, IL-8, and TNF-α, we have presented those results in addition to the other cytokines which were different in OA when compared to YA. To avoid duplicating the results, all other cytokine data are presented in text on lines 204-222.
Lines 199-200: “. There were also no significant changes across time points for IL-10 (p = 0.468)”. However, in lines 322-323 the authors described “anti-inflammatory IL-10 decreased at both 3 and 6 hours compared to baseline in the current study”. Please, correct it.
Thank you for your attention to detail. The results were correct, and we have reworked the discussion for clarity. The revised manuscript states the following on lines 335-343
“Interestingly, in the current study anti-inflammatory IL-10 decreased at 3 hours post-meal compared to baseline. This is consistent with prior work during the post-prandial period [22]. It has been suggested that the decrease in IL-10 may be explained by declines in IL-6, though the decrease in IL-10 was not significant in the present study. Although IL-6 may be released as a pro-inflammatory cytokine related to increased risk of CVD[33], it can serve as an anti-inflammatory myokine secreted from skeletal muscle with exercise [34]. Considering IL-6 upregulation with exercise may downstream increase IL-10 [35], the decrease in IL-6 observed here in the present study may partially explain the lack of any changes in anti-inflammatory IL-10.”

Reviewer 3 Report
The present study hypothesizes that aging would elevate systemic and airway inflammation, and analyzes whether acute exercise could reduce systemic and airway inflammation in older vs. younger adults. This is a study with a complete experimental design that articulates the effects of diet, physical exercise, and inflammatory mechanisms, providing answers to the molecular mechanisms that occur during these processes and possible association with different pathologies.
However, the authors should take into account the following observations for publication:
Please improve the wording of the abstract, so that it is better understood.
The introduction lacks a definition of the justification for the study; it is not based on a public health context that leads to the performance of this type of study, as in this case, cardiovascular diseases.
The sample size is small.
Describe the inclusion criteria of the participants, for example, whether they had any pathology.
Correct the y-axis in Figures 2 and 3.
In this type of study, in which postprandial states are evaluated, it is recommended to perform biochemical tests such as lipids and glycemia.
Separate the analyses by sex.
Update bibliographic references.
Author Response
Please improve the wording of the abstract, so that it is better understood.
We have revised the abstract for clarity and fixed several grammatical errors.
The introduction lacks a definition of the justification for the study; it is not based on a public health context that leads to the performance of this type of study, as in this case, cardiovascular diseases.
We have revised the introduction and discussion to clarify the importance of using post-prandial meal challenges for cardiovascular disease risk. This explanation reads as follows in the introduction and discussion on lines 33-43 and 280-284, respectively,
“The typical western diet consists of nutrient-poor and calorically dense foods, that are high in saturated and trans fats as well as sugar. In addition, individuals may consume multiple meals a day, placing them in a post-prandial state for most of the day [1]. The negative health effects of chronic high-fat meals (HFM) span from metabolic dysfunction to pulmonary complications that include but are not limited to: obesity, type 2 diabetes, cardiovascular disease (CVD), and respiratory diseases such as asthma [2, 3]. A key mechanism for the development of such lifestyle-related diseases is the activation of the immune system with persistent low-grade inflammation [4]. While HFM diets are associated with chronic low-grade inflammation, even an acute HFM meal increases systemic and airway inflammation [5, 6], though results are conflicting [7–9]. Considering inflammation is a hallmark characteristic of CVD, identifying who may be most susceptible to deleterious changes after meal consumption and how to best treat inflammation are paramount for improving medical care”.
“Determining who may be most susceptible to changes in inflammation is critical since inflammation is present in pulmonary and metabolic aliments and CVD. Also, identifying whether exercise may potentially mitigate post-prandial inflammation and ultimately chronic disease risk is an important area for clinical application and possibly treatment”.
The sample size is small.
We have addressed this concern on lines 395-396 in the experimental considerations per the Editor’s and reviewer’s request.
Describe the inclusion criteria of the participants, for example, whether they had any pathology.
While we have defined the participant characteristics in our previous work, we have also briefly included the inclusion criteria on lines 73-74 which state,
“The participants were free of CVD, type 2 diabetes or pulmonary diseases and were not on any medications known to interfere with any outcomes assessed in the present study.”.
Correct the y-axis in Figures 2 and 3.
The axes are corrected now.
In this type of study, in which postprandial states are evaluated, it is recommended to perform biochemical tests such as lipids and glycemia.
In the manuscript we mentioned that these data are from a larger study, which evaluated post-prandial lipemia, glycemia and metabolic load index (Kurti et al., 2021, British Journal of Nutrition). This is found on lines 68-70.
Separate the analyses by sex.
Please see our response to the Editor. However, we agree this is important and have included the following to the experimental considerations on lines 394-395
“Our goal was to match the groups by sex, however, enrolling OA who met the inclusion criteria was challenging. While the small sample size could have limited our ability to assess on our main outcomes, sub-analyses (data not shown) show no sex by time interactions for eNO or inflammation. Thus, sex is unlikely to have influenced these findings”.
Update bibliographic references.
All references have been updated accordingly.

Round 2
Reviewer 2 Report
The authors addressed all my early concerns.